# Characterization of biliary microbiota dysbiosis in extrahepatic cholangiocarcinoma

**Massa Saab** [1]*, **Denis Mestivier**[1,2], **Masoudreza Sohrabi**[3], **Christophe Rodriguez**[2,3,4], **Mahmood Reza Khonsari**[3], **Amirhossein Faraji**[3], **Iradj Sobhani**[1,2,3,4,5]*

1 Early Detection of Colon Cancer Using Molecular and Microbial Markers (EC2M3)-EA 7375 Team, University Paris-Est Creteil (UPEC), Créteil, France, 2 Bioinformatic Core Facility, IMRB UMR 955 – INSERM, UPEC, Créteil Cedex, France, 3 GastroIntestinal and Liver Diseases Research Center (GILDRC), Iran University of Medical Sciences and Firouzgar hospital (Prof. F. Zamani), Tehran, Iran, 4 Team 8, Inserm U955, Créteil, France, 5 Department of Gastroenterology, CHU Henri Mondor-APHP, Créteil, France

* massa.saab93@gmail.com (SM); iradj.sobhani@aphp.fr (SI)

**Data Availability Statement:** The 16S rRNA amplicon sequencing data are available from the European Nucleotide Archive (ENA) database

## Abstract

Extrahepatic cholangiocarcinoma (CCA) accounts for 3% of digestive cancers. The role of biliary microbiota as an environment-related modulator has been scarcely investigated in CCA, and the putative impact of associated diseases has not been yet assessed. We characterized the biliary microbiota in CCA patients in order to identify a specific CCA-related dysbiosis. The biliary effluents were collected through an endoscopic retrograde pancreatic cholangiography (ERCP) examination involving 28 CCA and 47 patients with gallstones, herein considered as controls. The biliary effluents were submitted to bacterial DNA extraction and 16S rRNA sequencing, using Illumina technology. Overall, 32% of CCA and 22% of controls displayed another associated disease, such as diabetes, pancreatitis, inflammatory bowel disease, or primary sclerosing cholangitis. Such associated diseases were considered in the comparisons that were made. Principal coordinate analysis (PCoA) detected a significant disparity of biliary microbiota composition between CCA patients and controls without an associated disease. Amongst the most abundant phyla, *Proteobacteria* did not significantly differ between CCA patients and controls, whereas *Firmicutes* levels were lower and *Bacteroidetes* higher in CCAs' biliary microbiota than in the controls' microbiota. The most abundant genera were *Enterococcus*, *Streptococcus*, *Bacteroides*, *Klebsiella*, and *Pyramidobacter* in CCA's biliary microbiota. Additionally, levels of *Bacteroides*, *Geobacillus*, *Meiothermus*, and *Anoxybacillus* genera were significantly higher in CCA patients' biliary microbiota, without an associated disease, in comparison with controls. A specific CCA-related dysbiosis was identified as compared to controls independently from associated diseases. This suggests that a microorganism community may be involved in CCA pathogenesis.

## Introduction

Cholangiocarcinomas (CCA) are relatively rare malignant tumors of the bile ducts, representing around 3% of digestive cancers [1]. CCA are either intrahepatic (CCAi) or located only in the extrahepatic biliary tracts (CCAe). CCAi constitutes the second-most common malignant

(http://www.ebi.ac.uk/ena) under the accession number PRJEB43183.

**Funding:** Funding for DNA analyses: UFR Sante-Paris12; UPEC Creteil-France; this routine annual funding of EC2M3 Lab had no role in study design, data collection and analysis, decision to publish, or preparation of the manuscript.

**Competing interests:** No Competing Interest.

liver tumor, representing 10–20% of primary liver cancers, following hepatocarcinomas that are the most common. In France, about 2,000 new cases are registered per year, resulting in CCA being the sixth-most-common malignant tumor of the gastrointestinal tract, whose incidence is increasing. In several Asian countries including the middle east, the incidence of CCA is still higher than in Western countries [2].

The circumstances of discovering a CCA differ according to the location. The diagnosis is often delayed, thus only made at an advanced disease stage [3, 4]. The endoscopic retrograde cholangiopancreatography (ERCP) of pancreato-biliary tracts is an invasive technique, which is nonetheless necessary for both CCA diagnosis and therapy. This investigation enables the collection of biological effluents like the bile on which the cytology analysis is conducted. This analysis is designed to confirm CCA diagnosis. The prognosis of biliary tumors is poor, with a 5-year survival rate estimated at 5%. The surgery is currently considered the standard of curative therapy [5]. To improve prognosis of CCA, its pathogenesis needs deeper investigation. This is challenging, given that CCA is a heterogeneous disorder that is likely influenced by environmental factors [6]. Thus, one could assume that, like for many digestive cancers whose incidence has increased during recent decades due to environmental factors, the CCA incidence would continue to rise.

Ulcerative colitis and primary sclerosing cholangitis (PSC) are two benign diseases that both enhance the risk of developing CCA in the Western world [7]. These diseases induce a chronic inflammation within the bile ducts that favors carcinogenesis. Intrahepatic biliary lithiasis, a common disease in Asia, is another inflammatory disease that enhances the risk of CCA. In Southeast Asia, CCA is associated with parasitic infections like *Opistorchis viverrini* and, albeit less well-established, with *Clonorchis Sinensis* infections [8].

Although exhaustive analysis of environment factors is challenging, this has been facilitated by studying the microbiota [9–12]. Indeed, changes within the human microbiota, which are roughly designed as dysbiosis, have been associated with the development of colon, oral, and lung cancers [10–12]. Microbiota is thought to generate physiological function alterations, which are likely to promote different diseases, including various cancers [13].

To illustrate, an increase in short-chain fatty acid producers, through changes in the numbers of *Fusobacterium*, *Prevotella*, and *Campylobacter*, has been observed in colorectal cancer patients [14]. Notably, *Fusobacterium*, *Prevotella*, and *Campylobacter* were all three found to be significantly more abundant in CCA, further suggesting they may play a role in gastrointestinal cancers, possibly through unregulated inflammatory response [15]. However, the distinct features of gut microbiota in patients with CCA and the interactions of CCA with gut microbiota have not yet been clarified. *Xiaodong et al.* were the first to characterize intestinal microbiota, bile acids, and cytokines in patients with CCA. These authors proposed some diagnostic markers, including short chain acids of biliary acid metabolites and two bacteria genera (*Lactobacillus* and *Alloscardovia*) for identifying CCAi [16].

Interestingly, microorganisms may also be detected in organs (*e.g.*, human female reproductive tract and human urinary system) whose effluents are assumed to be sterile [17, 18].

The bile, whose main components include bile acids, cholesterol, and phospholipids, functions as a biological detergent that emulsifies and solubilizes lipids, thus conferring them antimicrobial activity [19]. For this reason, the healthy biliary tract is generally considered a sterile environment [20]. However, based on metagenomics sequences, recent human [21] and animal studies [22] have indicated that the gallbladder may also harbor a complex of non-cultivable bacteria under non-pathological conditions [20].

Furthermore, sequencing of 16S ribosomal ribonucleic acid (RNA) has rendered it possible to estimate taxonomic profiling according to phylum, class, order, family, or genus. Moreover, assignment into species and sub-species generally requires whole metagenomic sequencing

[23]. However, up to now, biliary microbiota characterization remains fragmentary. We have thus planned to investigate, in an Asian medical center within a high CCA incidence area, a series of 30 extrahepatic CCA patients who underwent ERCP, in an effort to identify the biliary microbiota signature.

Accordingly, consecutive patients being referred for endoscopic retrograde cholangiopancreatography (ERCP) and exhibiting stigmata of CCAe were considered for analysis, under the condition that CCAe was proven using cytological or histological exams. Results were compared to those of a series of patients affected with benign biliary diseases (PBBs), who were referred to the same endoscopic unit for ERCP during the same period. These patients were considered as controls. Overall, 16S RNA sequencing demonstrated significant differences in the biliary microbiota composition between the two populations, which could implicate CCA-associated dysbiosis playing a role in biliary carcinogenesis.

## Population, materials, and methods

**Patients & materials.** Between 2014 and 2016, biliary fluid was collected during ERCP from 75 consecutive individuals with biliary obstruction, who were referred to tertiary centers.

In line with the Declaration of Helsinki, the research protocol of surveillance and of obtaining blood and bile samples of patients suspected of having either CCA or PBBS (obviously biliary duct lithiasis) was approved by the Ethic Committee of Iran University of medical Sciences under ID: IR.IUMS.REC.1397.115.

Following full explanation of the study's aim and procedures by a physician, the patients signed a formal consent document, which is to be held by the Iranian partner for 15 years. All participants were adults with high-normal cognition. The inclusion criteria were as follows: adults of 21 years or more; suspected of biliary obstruction due to either cholangiocarcinoma or lithiasis based on elevated liver enzyme and total bilirubin levels; presence of stricture on endosonography or magnetic resonance cholangiopancreatography (MRCP). The exclusion criteria were as follows: history of viral hepatitis, metabolic hepatitis, autoimmune hepatitis, alcoholic hepatitis, non-alcoholic steatohepatitis (NASH), medication-induced hepatitis, and chemotherapy. The patients underwent ERCP and brushing. To avoid contact and contamination with the duodenal mucosa upon ERCP, once the endoscope canal exit was positioned to the biliary duct entry, a biliary catheter was used for bile duct canalization and a 1-2mL of bile collected before brushing for cytology; both samples were used for microbiology and pathology analyses and were transferred to the Gastrointestinal and liver research center (GILDRC) Lab and conserved at -80˚c until analysis. Those patients with cytology or biopsies suggestive of CCAe were enrolled until the lesions were histologically confirmed based on biopsy or surgical (if applicable) samples. During the study period that did not exceed 15 months, the whole study population comprised 30 CCAe patients and 50 patients with a lithiasis, the latter being considered as controls, due to the final diagnosis of benign biliary pathology. Five patients were excluded from analysis (two in the CCA and three in the control groups) because of missing sample or disease confirmation. The frozen samples were transferred to the EC2M laboratory in France for deoxyribonucleic acid (DNA) extraction and sequencing. Briefly, 28 CCAe patients were compared to 47 controls. Patients were categorized into two additional subgroups: those with associated diseases, such as diabetes, inflammatory bowel disease (IBD), primary sclerosing cholangitis (PSC), and pancreatitis, as well as those without any associated diseases.

## Methods

**DNA extraction.** Biliary juices were immediately frozen after ERCP until DNA extraction. An unbiased DNA extraction procedure was applied to all samples, before targeted

metagenomics was performed. Essentially, pre-extraction by homogenization of the beads, which is associated with cell rupture, was followed by extraction using QiaSymphony (Qiagen, Hilden, Germany), as described by commercial instructions. A negative control was tested in each series, with positive controls used to evaluate the performance of metagenomic techniques for detecting bacteria Zymobiomics®, Ozyme, Montigny le Bretonneux, France).

**Targeted metagenomics.** Targeted metagenomics (TM) comprised the study of amplicon libraries through the V3-V4 (16S-V3V4) domains of the 16S rRNA bacterial gene [24]. The amplicon was prepared from 5mL of extracts, according to the 16S Metagenomic Sequencing Library Preparation Protocol that was provided by the manufacturer (Illumina, San Diego, California, USA). Quality control was assessed using a D1000 ScreenTape on a TapeStation (Agilent, Santa Clara, California, USA), and quantity was determined via the Quant-it dsDNA Assay kit (ThermoFischer, Waltham, MA, USA) on a Mithras LB 940 (Berthold Technologies, Bad Wildbad, Germany).

All libraries were normalized to nM, pooled, and denatured before end-of-pair sequencing (v3-v4, 2 x 300 bp) on a MiSeq device (Illumina, San Diego, California, USA). The targeted bacterial regions were sequenced according to the manufacturer's instructions [25]. The 16S rRNA amplicon sequencing data are available from the European Nucleotide Archive (ENA) database (http://www.ebi.ac.uk/ena) under the accession number PRJEB43183.

Overall, 10,483,756 paired-reads were sequenced (137,944.16 +/- 41,883.18 reads/sample, length: 25-251bp). After quality checking with the FastQC (https://www.bioinformatics.babraham.ac.uk/projects/fastqc/, v0.11.9) software, reads were filtered for quality that was less than Q20 (using a sliding window of 5pb) and minimal length of 100bp (using trimmomatic, v0.39) [26]. Remaining quality paired-end reads were merged using the FLASH2 software (v2.2.0, https://github.com/dstreett/FLASH2/blob/master/README) [27], yielding 120,716.70 +/- 38,924.65 (87.4 +/- 5.6%) combined paired per sample. We then employed MALT (v0.4.1, https://software-ab.informatik.uni-tuebingen.de/download/malt/welcome.html) and Megan6 (v6.17.0) [28] software with the rRNA SILVA database (v132) [29] for taxonomical assignation, using parameters for 16S ("SemiGlobal" / "mif" / "LCA"). Abundances were exported as a Biological Observation Matrix (BIOM) for further analyses (S1 File).

**Data analysis.** Gender, associated diseases, and diabetes differences were assessed using Chi-squared testing (GraphPrism 8), while age and body mass index (BMI) differences were assessed via t-testing (normal distribution was assessed using the D'Agostino-Pearson test). Analyses were performed including age and BMI of patients as co variable. Values were considered statistically significant when $p < 0.05$. PCoA, figures, and differential microbiota analysis were performed using the Shiny Application for Metagenomic Analysis (Shaman), from *Institut Pasteur de Paris* (http://shaman.pasteur.fr/) [30]. For the differential abundance analysis, we used the BIOM Table and the "target" File that associates each sample with its explanatory variables (see suppl data: S1 File and S3 Table). Abundances of bacteria were summarized at the Phylum and Genus levels. The statistical analysis in SHAMAN is based on the DESeq2/R package which model abundance counts with a negative binomial distribution. For the experimental design, we selected the variables of gender, BMI, age, diagnosis (Control/CCA), and associated diseases. We also included the interaction between diagnosis and associated diseases in the model. We used defaults parameters, such as the "weighted non-null normalization," which was introduced by Volant et al [30] and accounts more accurately for matrix sparsity. SHAMAN/DESeq2 yields baseMean and FoldChange (and log2FoldChange) and an adjusted p-value. Outputs were analyzed using Benjamini-Hochberg adjustment method. To be retained as differentially abundant, a taxon (phylum or genus) had to fulfill the following criteria: p value adjusted $<0.05$ and baseMean $>100$.

# Results

## Patient characteristics

Overall, 75 patients with biliary obstruction were included in the analysis (n = 28 CCA and n = 47 controls). The characteristics of the patients are summarized in Table 1.

There was no significant difference between these two groups in terms of age, gender, or BMI. However, the number of associated diseases (diabetes, IBD, PSC, and chronic pancreatitis) differed between both groups, amounting to 22% in controls and 32% in CCA cases with at least one associated disease, «without reaching statistical significance". Consequently, these patients were reclassified into two subgroups for deeper analysis: those with associated diseases versus those without.

## Comparison of the most abundant genera and phyla

As the most abundant genera in the whole population (cases and controls), *Enterococcus*, *Streptococcus*, *Bacteroides*, *Klebsiella*, *Clostriduim*, *Fusobacteruim*, and *Pyramidobacter* were identified. Furthermore, a trend towards higher abundances of *Streptococcus*, *Bacteroides*, and *Pyramidobacter* in CCA cases contrasted with a trend towards higher abundances of *Clostriduim*, *Klebsiella*, *Fusobacteruim*, and *Enterococcus* in controls (Fig 1A).

Four genera, namely *Streptococcus*, *Enterococcus*, *Klebsiella*, and *Pyramidobacter*, were more abundant in CCA cases with associated diseases (n = 9), compared to those without (n = 19). The *Enterococcus* genus was most abundant in controls with associated diseases (n = 10), compared to those without (n = 37) (Fig 1B).

At the phylum level, *Proteobacteria* was the most abundant, in both cases and controls. In addition, *Firmicutes* were more abundant in controls (particularly in those with associated co-morbidities), while *Bacteroidetes* were more abundant in CCA cases (Fig 2A and 2B).

## Comparison using principal coordinate analysis (PCoA)

Analysis of the entire biliary microbiota failed to show any significant differences between cases and controls, but there was a trend towards between-group differences (p-0.058,

**Table 1. Characteristics of patients with cholangiocarcinoma (CCA) and controls.**

|  | Control N = 47 | CCA N = 28 | p value |
|---|---|---|---|
| Female, n (%) | 24 (51) | 9 (32) | 0.15 |
| Male, n (%) | 23 (49) | 19 (68) | 0.15 |
| Body mass index (BMI), mean (SD) | 27 (4) | 25 (5) | 0.18 |
| Age, mean (SD) Yr | 57 (17) | 64 (12) | 0.07 |
| Associated diseases, n (%) | 10 (22) | 9 (32) | 0.2 |
| Diabetes, n (%) | 9 (90) | 6 (67) | 0.2 |
| Pancreatitis, n (%) | 1 (10) | 0 | ND |
| Inflammatory bowel disease (IBD), n (%) | 0 | 2 (22) | ND |
| Primary sclerosing cholangitis (PSC), n (%) | 0 | 1 (8) | ND |
| Tumor differentiation, n (%) | - | 28 (100) | ND |
| Grade 1*, n (%) | - | 2 (7) | ND |
| Grade 2*, n (%) | - | 8 (29) | ND |
| Grade 3*, n (%) | - | 6 (21) | ND |
| Grade 4*, n (%) | - | 12 (43) | ND |

Comparisons were performed by using Chi-squared tests for qualitative parameters and t-test for quantitative parameters. ND: not determined.

*according to the International TNM classification.

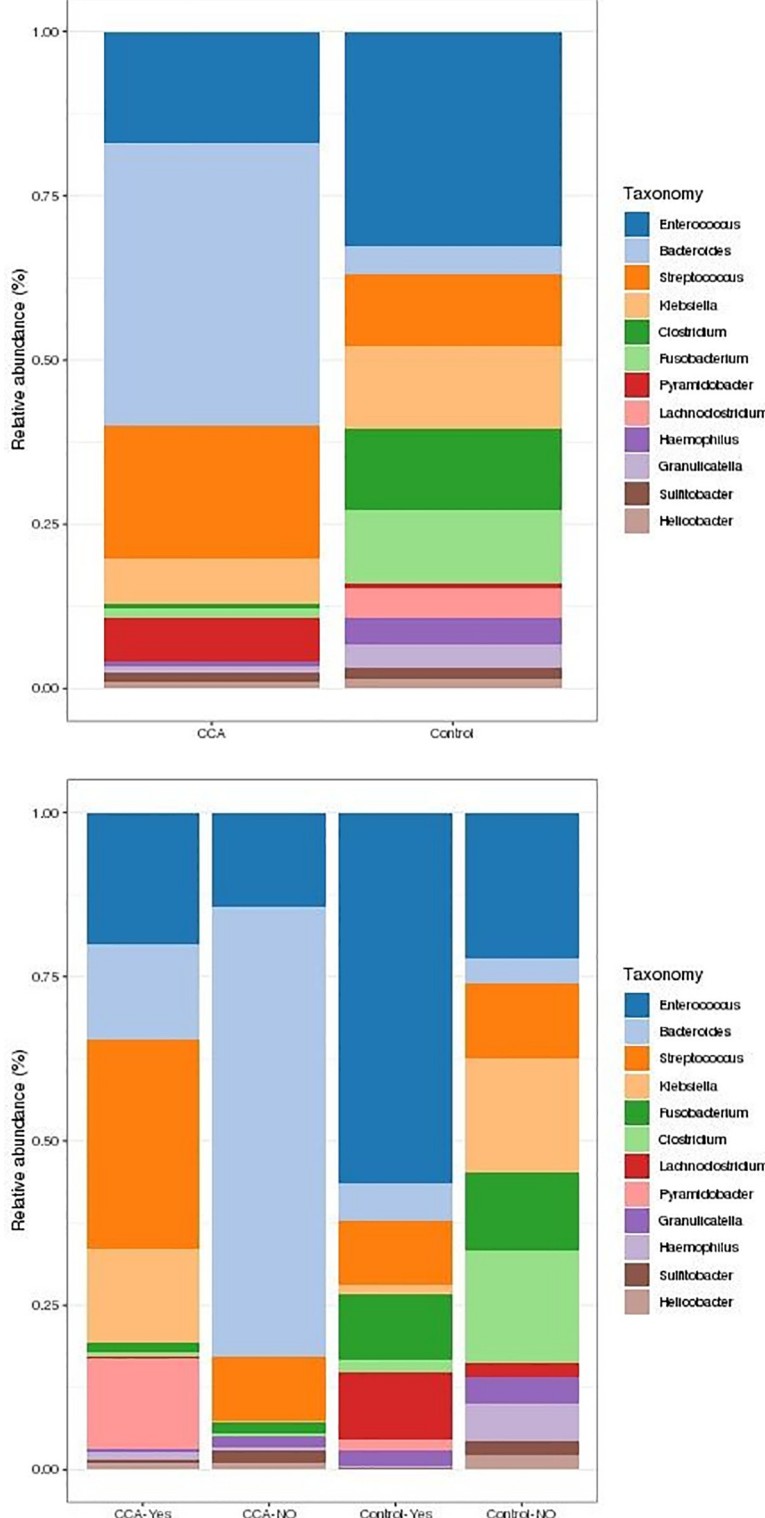

**Fig 1. Relative abundances (%) of genera in patients and controls.** The 12 most abundant genera are illustrated in the cases and controls. **A)** CCA (n = 28) and controls (n = 47), regardless of associated diseases. **B)** CCA and controls, with (Yes) or without (NO) associated diseases.

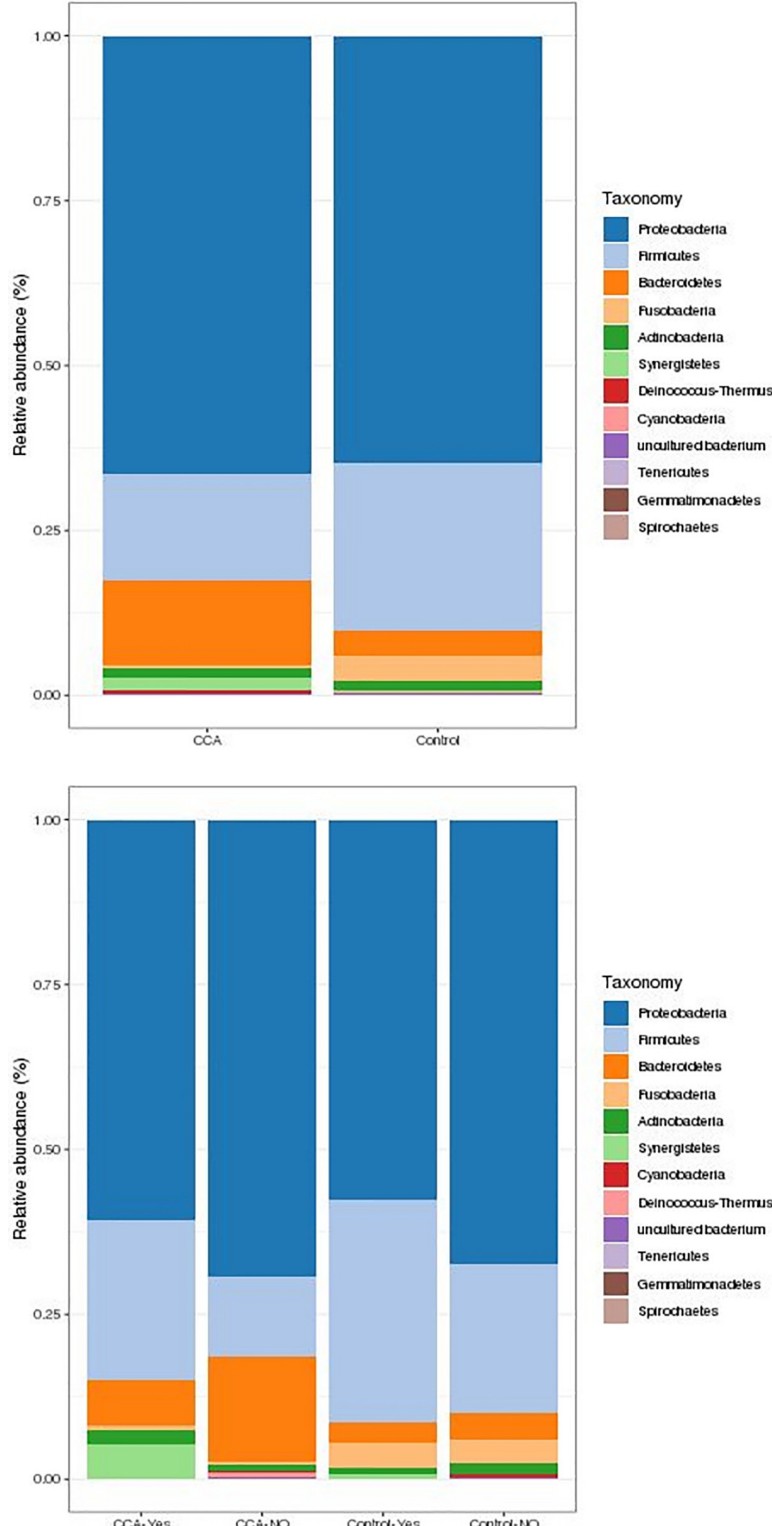

**Fig 2. Relative abundances (%) of phyla in patients and controls.** The 12 most abundant phyla in all individuals. **A)** CCA (n = 28) and controls (n = 47), regardless of associated diseases. **B)** CCA and controls, with (Yes) or without (NO) associated diseases.

Permanova test) (Fig 3A). However, when these analyses were limited to associated disease-free cases and controls, a significant separation was observed (p <0.01, permanova test) (Fig 3B). This difference was unaffected by age, gender, or BMI.

### Differential bacteria in biliary microbiota (CCA cases versus controls)

The composition of biliary microbiota significantly differed between cases and controls. Several genera were significantly different between both groups (Table 2A; S1A Table). At phyla level, we observed that *Firmicutes*, *Fusobacteria*, and *Cyanobacteria* levels were significantly lower in CCA cases than controls (Table 2B and S1B Table).

When analysis was limited to those individuals without associated diseases (19 CCA cases versus 37 controls), we identified 11 bacterial genera that significantly differed between cases and controls (Table 3A). The genera levels of *Bacteroides*, *Geobacillus*, *Meiothermus*, and *Anoxybacillus* were significantly higher in cases versus controls (Table 3A and S2A Table). Similarly, at the phylum levels, the levels of *Deinococcus-Thermus* were higher and those of *Firmicutes*, *Fusobacteria*, and *Actinobacteria* were all lower in CCA cases than controls (Table 3B and S2B Table).

## Discussion

Environmental factors play a role in biliary carcinogenesis [6]. To better understand this role, we characterized biliary microbiota in a group of patients that were suffering from CCA. By using a targeted amplicon sequencing approach for 16s rRNA, we could thereby show that the bacteria composition significantly differed in CCA, compared to individuals without CCA that had undergone ERCP for bile collection. Some of the subjects, in either controls or cases, suffered from associated diseases, such as diabetes, pancreatitis, PSC, and IBD, too. All these diseases are known influencing the gut microbiota composition. Thus, we analyzed a subgroup of CCAe cases in whom biliary juice could be obtained through endoscopy exam in comparison with controls, none of which exhibited inflammatory or neoplastic associated diseases. Consequently, we could identify various genera, the abundances of which were shown to vary in cases as compared to controls, strongly suggesting CCAe being linked to dysbiosis.

To the best of our knowledge, previous studies that have demonstrated an association between biliary microbiota dysbiosis and human diseases have mainly concerned the process of cholesterol gallstone formation [22, 23]. These data suggested human gallbladder microbiome possibly playing a physiological role and influencing biliary metabolic profile. In a recent study conducted in patients with cholesterol gallstones [22], the connection of bile bacteria population and gut microbiota has been analyzed. In this paper, the authors argue high microbial diversity in the bile duct being impacted by intestinal microbiota diversity. Whether the differential bacteria panel we found in the biliary fluid is specifically linked with biliary carcinogenesis should be further discussed.

In this pilot study, our aim was to verify that biliary microbiota dysbiosis may be a key contributor to biliary neoplasia. Previous studies have shown that bile fluid dysbiosis could be linked to various diseases, including biliary lithiasis [21, 31–33]. This latter condition induces partial or total obstruction of biliary flux. The point that PCoA analysis revealed a significant separation between CCAe and lithiasis subgroups may suggest that biliary microbiota changes could be favored by reduced biliary flow. However, this hypothesis seems unlikely, although bacteria may find a more favorable environment to growth in obstructed bile ducts. By collecting bile fluids above the occlusion using the ERCP approach, we found that patients referred for lithiasis and CCAe can be considered comparable with respect to this bias. Furthermore, consistent with previous studies [21, 34–38], we found that phyla (Table 2B and S1B Table)

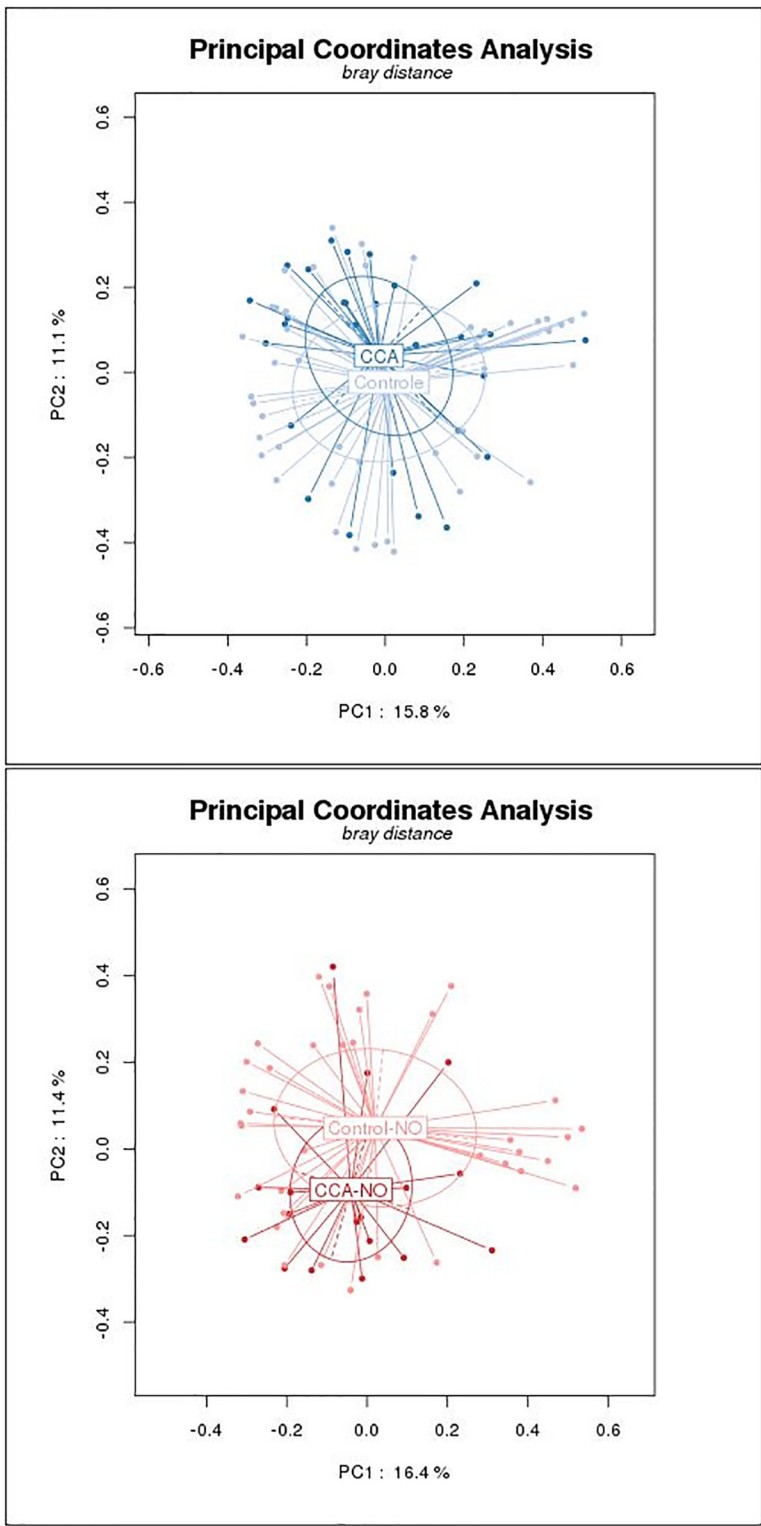

**Fig 3. Principal coordinate analysis (P CoA) of biliary microbiota in patients and controls. A)** PCoA according to the diagnosis, in all individuals (permanova test; p = 0.058). **B)** PCoA according to the diagnosis, limited to patients without associated diseases (permanova test = 0.01).

**Table 2. Significant differential abundances, according to the diagnosis (CCA *vs*. controls).**

A.

| Genus | Base mean | Fold Change | p value_adjusted |
|---|---|---|---|
| *Clostridium* | 2597.72 | 0.10 | 0.0031 |
| *Fusobacterium* | 2455.33 | 0.06 | 0.0005 |
| *Granulicatella* | 800.17 | 0.12 | 0.0027 |
| *Pseudomonas* | 715.27 | 0.12 | 0.0018 |
| *Bacillus* | 468.05 | 0.26 | 0.0488 |
| *Actinomyces* | 252.48 | 0.18 | 0.0276 |
| *Citrobacter* | 129.8 | 0.05 | 0.0005 |
| *Campylobacter* | 104.23 | 0.20 | 0.047 |

B.

| Phyla | Base mean | FoldChange | pvalue_adjusted |
|---|---|---|---|
| *Firmicutes* | 22438.7 | 0.48 | 0.0491 |
| *Fusobacteria* | 2713.32 | 0.10 | 3.37E-05 |
| *Cyanobacteria* | 272.55 | 0.17 | 0.0111 |

The list of bacteria include those genera at the mean base >100 (mean base of the whole cohort). Comparisons (CCA versus controls) yielded fold changes that are expressed in the values of ratios of the cases/controls; Statistical analysis using the DESeq2 R package. A) Genus level. B) Phylum level.

*Proteobacteria*, *Firmicutes*, *Bacteroidetes*, *Fusobacteria*, and *Actinobacteria* dominated the bile microbiota in our population. Some genera or species, such as *Helicobacter pylori*, have been found to be present at higher levels in CCA [15] than in controls. Yet, this was not confirmed in our series, suggesting that *Helicobacter pylori* presence may suggest stomach juice-induced

**Table 3. Significant differential abundances according to the diagnosis (CCA *vs*. controls).**

A.

| Genus | Base mean | Fold Change | p value_adjusted |
|---|---|---|---|
| *Bacteroides* | 4388.85 | 7.06 | 0.0304 |
| *Klebsiella* | 3125.04 | 0.02 | 8.95E-08 |
| *Clostridium* | 2597.72 | 0.01 | 1.17E-09 |
| *Fusobacterium* | 2455.33 | 0.06 | 0.0007 |
| *Haemophilus* | 892.5 | 0.05 | 7.61E-06 |
| *Enterobacter* | 569.44 | 0.06 | 0.0009 |
| *Geobacillus* | 530.19 | 7.95 | 0.0067 |
| *Actinomyces* | 252.48 | 0.20 | 0.0377 |
| *Meiothermus* | 199.74 | 52.05 | 5.19E-06 |
| *Anoxybacillus* | 190.83 | 10.66 | 0.0378 |
| *Citrobacter* | 129.8 | 0.03 | 6.16E-05 |

B.

| Phyla | Base Mean | Fold Change | p value_adjusted |
|---|---|---|---|
| *Firmicutes* | 22438.7 | 0.38 | 0.0054 |
| *Fusobacteria* | 2713.32 | 0.08 | 9.98E-05 |
| *Actinobacteria* | 1505.03 | 0.38 | 0.0490 |
| *Deinococcus-Thermus* | 202.1 | 19.53 | 2.10E-07 |

The list of significantly differential bacteria between cases and controls include the mean base >100. Comparisons (n = 19 cases versus n = 37 controls, without associated co-morbidities) yielded fold changes that are expressed in values of case/control ratios. Statistical analysis using the DESeq2 R package. A) Genus. B) Phylum.

contamination. In addition, *Proteobacteria* is presumed to represent about 30% of stomach phyla and 60% of bile duct phyla [39]. Notably, the levels of *Proteobacteria* found in our series seems to be close to the values described in the small intestine by other authors [40]. These discrepancies in biliary microbiota composition may be accounted for by various conditions including gastric or duodenal contamination. It must be mentioned that although we took specific precaution in collecting biliary fluid (see methods) in the current study to prevent biliary tract juice from being contaminated by gastric or duodenal juices, it is not possible to rule out this bias, as additional effluent samples from stomach, duodenum and intestines have not been cross checked in our series. Amongst other potential factors that may affect bacteria colonization and survival within the biliary tract, the sphincter of Oddi function close to the tumour location could have been altered leading to increased inner pressure of the proximal duct. Alternatively, the pipelines used for taxonomic assignment that are known to vary from study to study could have impacted bacteria assignation. Nevertheless, despite all these biases, it can be assumed that differences in either intestinal or biliary microbiota composition of CCA patients do illustrate different bacteria environments that are associated with neoplasia, in comparison with individuals without tumors. According to this hypothesis, *Jia et al* [16] analyzed fecal microbiota in a series of intra hepatic CCAi, presumably without any contact with intestinal microbiota. These authors revealed that bacteria communities including *Lactobacillus*, *Actinomyces*, *Peptostroptococcacae* and *Alloscardovia* were found to be more abundant in gut microbiota from cases that controls. Thus, if involved in carcinogenesis, bacteria could act not only as adherent cells to the biliary tissue but also through the enterohepatic metabolic cycle of microbiota, such as biliary acid metabolism. This pathway has also been suggested to explain tumor growth and outcomes through host immune response to CCA [41]. Further studies including large series of both CCAe and CCAi patients are now required to more deeply characterize biliary cancer-associated dysbiosis as co carcinogenic and prognostic marker.

The dysbiosis we currently identified as being significantly linked to CCAe comprised genera, such as *Bacteroides*, *Geobacillus*, *Anoxybacillus*, and *Meiothermus*, which were found more abundant in cases than in controls.

*Bacteroides* are Gram-negative, strict anaerobic, non-spore-forming bacilli, which are intestinal microbiota bacteria. Several studies have demonstrated associations between *Bacteroides* and colon cancer [42, 43]. In addition, elevated *Bacteroides* have been found in various other diseases and conditions, including arthritis in transgenic rats HLA-B27 [44, 45]. Studies in germ-free mice have shown that the *Bacteroides* antigen contributes to the recruitment and proliferation of low-avidity CD8+ T lymphocytes; these cells may be similar to thymic CD4 + Tregs, as well as to the response to chronic antigenic exposure in intestinal lymphoid tissues [46]. The enrichment of *Bacteroides* in arthritis patients likely indicates that this condition may contribute to disease progression. The expansion of *Bacteroides* has been hypothesized to be a compensatory mechanism for regulating autoimmune reactions. Moreover, higher abundance of *Bacteroides* in the bile has been linked to cholethiasis through metabolomic changes [47]. Its role via auto-mmune disease mechanisms cannot yet been ruled out. Whether these bacteria are involved in primary cholangitis and constitute a marker of auto-immune disease has been investigated. Controversial results on PSC-associated biliary microbiota have been published: Pereira *et al* [48] failed to find any significant association between PSC and biliary microbiota changes, whereas a specific gut microbiota dysbiosis was characterized [42]. We were not in the position to verify this specific point, because only one of our CCA patients suffered from PSC. Regarding the potential association of bile bacteria and biliary CCA, two other studies have similarly employed 16S rRNA sequencing to characterize biliary tract

bacteria [15, 49]. The impact of biliary bacteria composition on patients' outcome has not been demonstrated. Despite a trend towards biliary dysbiosis variations according to stages and one-year survival rates, in our series, we were unable to identify any significant microbial predictive markers able to assess CCA patients with bad prognosis (data not shown). This is most likely due to the small size of our series.

To our knowledge, *Geobacillus*, *Meiothermus*, and *Anoxybacillus* levels, which were found to be more abundant in CCAe, have not previously been linked to CCA. The genus *Geobacillus* comprises a group of Gram-positive thermophilic bacteria, which are able to grow in an anaerobic milieu above the range of 45–75˚C [50]. The genus *Meiothermus* is a thermophilic environmental bacteria, which is isolated upon a hot spring [51]. The genus *Anoxybacillus* is a rod-shaped bacterium from the *Bacillaceae* family, which forms spores that are likely to resist the geothermal spring milieu [52]. Whether these associations illustrate an accompanied bacteria community rather than a causative core cannot be addressed by our study. Interestingly, a recent study has reported an increase in *Anoxybacillus* and *Geobacillus* genera in bladder cancer patients [53]. Further, two genera (*Lactobacillus* and *Alloscardovia*), have been shown to be associated with CCAi, suggesting that a possible metabolic pathway alteration could possibly favor biliary carcinogenesis [16].

Our study has some limitations. The biliary microbiota characterization has not yet been carried out in healthy subjects, owing to ethical and technical difficulties. We could not verify whether the bacteria we found within the biliary lumen were similar to those that are adherent to tumoral tissues. Indeed, many patients did not undergo surgical tumor resection. Although all precautions were taken to avoid intestinal milieu contamination during ERCP collection, we cannot rule that bacteria originating from the duodenum were included in the collected fluids, since separated milieus were not screened.

## Conclusion

We have characterized the biliary microbiota in CCAe patients and compared it with controls. To avoid confounding factors of associated diseases, we considered CCA patients and controls that were both free of co-morbidities. By doing so, we revealed significant changes in biliary microbial components. This suggests a significant CCA-associated biliary dysbiosis may enable us to distinguish these patients from non-cancerous controls. Although our results showed that the associated diseases do modify the composition of the biliary microbiota, we suggest that some of these bacteria may still be involved in CCA development.

## Supporting information

**S1 Table. Significant differential abundances, according to diagnosis (CCA versus controls).** The total lists of all bacteria including genera (CCA versus controls) after comparisons. These yielded fold changes that are expressed in the log values of ratios of the cases/controls. A) Genus level. B) Phylum level. Statistical analysis were based on the DESeq2 R package. (XLSX)

**S2 Table. Significant differential abundances, according to diagnosis.** (CCA versus controls). The total lists of all bacteria including genera (CCA versus controls) after comparisons between CCA cases (n = 19) and controls (n = 37), without associated co-morbidities. These yielded fold changes that are expressed in the log values of ratios of the cases/controls. A) Genus level. B) Phylum level. Statistical analysis were based on the DESeq2 R package. (XLSX)

**S3 Table. Target file.** Characteristics of the patients with cholangiocarcinoma and the controls.
(CSV)

**S1 File. Abundance table (BIOM format).** Number of reads associated with each taxon for each patient and taxonomy.
(XLSX)

## Acknowledgments

The authors would like to thank all the patients for accepting to contribute to the current research, in addition the employees of the Henri Mondor (Créteil, France) and Firouzgar (Tehran, Iran) hospitals, who were involved in the study. They also wish to thank the department Dean, Prof. Farhad Zamani, for supporting the study without any financial support from Iran to the French lab EC2M3. Finally, many thanks go to Vanessa Demontant for her technical help.

## Author Contributions

**Conceptualization:** Massa Saab.

**Formal analysis:** Denis Mestivier.

**Methodology:** Massa Saab, Christophe Rodriguez.

**Project administration:** Iradj Sobhani.

**Supervision:** Massa Saab, Iradj Sobhani.

**Validation:** Massa Saab, Denis Mestivier, Masoudreza Sohrabi, Mahmood Reza Khonsari, Amirhossein Faraji, Iradj Sobhani.

**Visualization:** Massa Saab, Denis Mestivier, Masoudreza Sohrabi, Christophe Rodriguez, Mahmood Reza Khonsari, Amirhossein Faraji, Iradj Sobhani.

**Writing – original draft:** Massa Saab, Denis Mestivier, Masoudreza Sohrabi, Christophe Rodriguez, Mahmood Reza Khonsari, Amirhossein Faraji, Iradj Sobhani.

**Writing – review & editing:** Massa Saab, Denis Mestivier, Masoudreza Sohrabi, Christophe Rodriguez, Mahmood Reza Khonsari, Amirhossein Faraji, Iradj Sobhani.

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
