## [Decision Letter · Decision Letter 0]

27 Aug 2020

PONE-D-20-20572

CHARACTERIZATION of biliary microbiota dysbiosis in extrahepatic cholangiocarcinoma

PLOS ONE

Dear Dr. Saab,

Thank you for submitting your manuscript to PLOS ONE. After careful consideration, we feel that it has merit but does not fully meet PLOS ONE’s publication criteria as it currently stands. Therefore, we invite you to submit a revised version of the manuscript that addresses the points raised during the review process.

There were major issues raised which related to lack of mechanistic details, small sample size and the mode of presentation and language issues.

Please submit your revised manuscript within 90 days. If you will need more time than this to complete your revisions, please reply to this message or contact the journal office at plosone@plos.org. Please include the following items when submitting your revised manuscript:

We look forward to receiving your revised manuscript.

Kind regards,

Pradeep Dudeja

Academic Editor

PLOS ONE

Journal Requirements:

2. Thank you for your ethics statement : 'All patients were enrolled at Firoozgar Hospital (Teheran) for the present translational study and provided their informed consent to conduct microbial analyses on the biliary juice recovered during the ERCP (Ethics Committee approval under ID: IR.IUMS.REC.1397.115 of Iran university of medical sciences).'

(a) Please amend your current ethics statement to confirm that your named institutional review board or ethics committee specifically approved this study.

(b) Once you have amended this/these statement(s) in the Methods section of the manuscript, please add the same text to the “Ethics Statement” field of the submission form (via “Edit Submission”).

For additional information about PLOS ONE ethical requirements for human subjects research, please refer to " ext-link-type="uri" xlink:type="simple">http://journals.plos.org/plosone/s/submission-guidelines#loc-human-subjects-research."

3. Please provide additional details regarding participant consent. In the ethics statement in the Methods and online submission information, please ensure that you have specified what type of consent you obtained (for instance, written or verbal, and if verbal, how it was documented and witnessed). If your study included minors, state whether you obtained consent from parents or guardians.

4. In your Methods section, please provide additional information about the participant recruitment method and the demographic details of your participants. Please ensure you have provided sufficient details to replicate the analyses such as: a) the recruitment date range (month and year), b) a description of any inclusion criteria that were applied to participant recruitment and c) a description of how participants were recruited.

5. Please provide a sample size and power calculation in the Methods, or discuss the reasons for not performing one before study initiation.

6. To comply with PLOS ONE submission guidelines, in your Methods section, please provide additional information regarding your statistical analyses. For more information on PLOS ONE's expectations for statistical reporting, please see https://journals.plos.org/plosone/s/submission-guidelines.#loc-statistical-reporting.

Reviewers' comments:

Reviewer's Responses to Questions

**Comments to the Author**

1. Is the manuscript technically sound, and do the data support the conclusions?

Reviewer #1: Yes

Reviewer #2: Partly

2. Has the statistical analysis been performed appropriately and rigorously? 

Reviewer #1: Yes

Reviewer #2: No

3. Have the authors made all data underlying the findings in their manuscript fully available?

Reviewer #1: No

Reviewer #2: Yes

4. Is the manuscript presented in an intelligible fashion and written in standard English?

Reviewer #1: Yes

Reviewer #2: No

5. Review Comments to the Author

Reviewer #1: • Major comments:

i. Data is held at institute and “may” be accessible under specific request is not adhering to the journals requirements to make all data underlying findings available online, especially since this is large data that could potentially lead to incredible findings in the fields of microbiome and CCA research.

ii. He authors claim no financial support for this work, however, this manuscript demonstrates intensive work.

iii. Wald test is performed for the differential abudance analysis, but due to small sample size 30 of CCA, it is recommended that likelihood ratio test be performed.

iv. There is no information on the severity of CCA in patients (or early/late stage PSC/IBD/etc in cases of co-morbidities) and there is no information on ethnicity or race on patients studied. The authors should attempt to present this information for transparency.

v. Methods in this manuscript should be applied to established data sets, especially those cited in the discussion, to verify that similar results are found. This manuscript claims to characterize CCA bile microbiome but does not verify its findings on publicly available data, or own previously published data sets. Also, there is no comparison of microbiome from CCA-No to those of other biliary diseases or duodenal samples which may demonstrate the true novel findings of this manuscript and reduce suspicion of collection method contamination.

• Minor comments:

i. All figures say “Non” instead of “No” when indicating comorbidity existence in CCA or control patients.

ii. Differential abundance should be presented in linear modeling graph for data presentation to reduce confusion of the base mean and log fold change.

iii. The discussion does not speculate the on the role Anoxybacillus, Geobacillus, and Meiothermus play in the biliary milieu. Additionally. There is no mention of bile altering bacteria levels which would be a great addition to this manuscript since bile flow, bile composition and cholestasis are interrelated in CCA and other biliary diseases.

iv. Introduction and discussion should be edited. There is limited information on the impact of biliary microbiome dysbiosis in CCA development or how important dysbiosis of microbiome composition is in other diseases. The detection method of CCA is not of high importance for the findings in this manuscript unless it is to address any potential contamination from method of collection (which should be reserved for discussion and not introduction).

v. Figures should be combined (all figure 1 together) rather than spread out. The figure legends should be placed outside of manuscript flow since that interrupts the readers comprehension.

Reviewer #2: The current manuscript submitted by Massa Saab et al was designed to characterize biliary microbiota in consecutive patients with a histologically proven CCAe, and their results were compared to a series of patients with benign biliary affected diseases (PBBs), which were considered the control. Using 16S RNA sequencing has demonstrated significant differences in the composition of the biliary microbiota between the two populations, which could implicate CCA-associated dysbiosis in the biliary carcinogenesis. Overall, 32% of CCA and 22% of control patients displayed another associated disease, such as diabetes, pancreatitis, inflammatory bowel disease, or primary sclerosing cholangitis. Comparisons considered associated diseases. Principal coordinate analysis (PCoA) detected a significant disparity of the biliary microbiota composition between the CCAs and controls without an associated disease. Levels of Bacteroides, Geobacillus, Meiothermus, and Anoxybacillus genera were significantly higher in CCA patients’ biliary microbiota, without an associated disease, than in the controls. A specific CCA-related dysbiosis is identified as compared to controls independently from associated diseases, therefore a microorganism community might be involved in the CCA pathogenesis, as suggested by authors. It is an interesting paper and the experiments were properly performed. However, the mechanisms between biliary microbiota and CCA development/progression are undefined, and the data on biliary carcinogenesis are descriptive. Additionally isolated CCA cells need to be better characterized, to make sure that they display the CCA phenotypes associated with biliary microbiota as expected. Moreover, the data is too preliminary and lack of the significance and mechanistic studies to define the specific biliary microbiota associated cancer signaling pathways.

Comments:

1. The major issue of the current manuscript is the mechanisms between biliary microbiota and CCA development/progression are undefined, and the data on biliary carcinogenesis are descriptive. Actually several manuscripts have proved the associations among gut microbiota, bile acid metabolism and cytokines in cholangiocarcinoma development and progression. Therefore further functional characterizations for CCA development and progression are need for the current manuscript.

2. The isolated/primary CCA cells from human CCA tissues were not well characterized and the authors failed to show the CCA phenotypes associated with biliary microbiota as expected. Therefore their cancer associated biliary microbiota phenotypes are questionable.

3. There is some confusion in the manuscript to prove that the levels of Bacteroides, Geobacillus, Meiothermus, and Anoxybacillus genera were significantly higher in CCA patients’ biliary microbiota, without an associated disease, than in the controls. More detailed mechanistic studies should be carried out to define the specific mechanisms of malignant transformation and their down stream signaling mechanisms.

4. The rationale should be provided in introduction. Summary is missing in the discussion section and the conclusion needs to be detailed.

5. Statistic analysis should be included in Figure 1 2 with some detailed CCA cancer stage and survival information.

6. There are not enough data in the main manuscript. Therefore the supplementary information should be moved to the main text.

7. English writing skills need to be improved. The format of the fonts and some technical terms should be consistent.

8. The discussion needs to be modified and focused. Some of the descriptions are disconnected from the central focus in the discussion section.

6. PLOS authors have the option to publish the peer review history of their article (what does this mean?). If published, this will include your full peer review and any attached files.

Reviewer #1: No

Reviewer #2: No

---

## [Author Response · Author response to Decision Letter 0]

25 Jan 2021

We would like to thank you for the consideration brought our paper, and for allowing us to resubmit a modified version of our manuscript to PLOS One. After careful revision, we responded in a “point-by-point manner” to your own comments as well as to the reviewers’ remarks. The new version has been revised by an English native lecture. We thank reviewers for their fruitful comments. One of reviewers recommended adding a mechanistic approach in the present study. We wish to stress out that several studies around main hypotheses are already in progress. Up to know there is no evidence which out of pathways we investigate is mainly involved in the carcinogenesis. This the reason why we decided including new data regarding tumor staging and long term follow up of patients. Unfortunately, we didn’t find any predictive value in bacterial changes for estimating prognosis likely due to the small size of our cohort cases. We are confident that our study needs to be reproduced through various ethnic populations before going more in-depth through one of these pathways.

We also would thank you for the additional time granted to prepare this present modified version. Consequently, the following items are considered in the revised manuscript:

• Responses to each point raised by the academic editor and reviewer(s). 

• A marked-up copy of our manuscript that highlights changes made to the original version and identified as 'Revised Manuscript with Track Changes'.

• An unmarked version of revised paper without tracked changes identified as ‘Manuscript'.

---

## [Decision Letter · Decision Letter 1]

15 Feb 2021

CHARACTERIZATION of biliary microbiota dysbiosis in extrahepatic cholangiocarcinoma

PONE-D-20-20572R1

Dear Dr. Saab,

We’re pleased to inform you that your manuscript has been judged scientifically suitable for publication and will be formally accepted for publication once it meets all outstanding technical requirements.

Kind regards,

Pradeep Dudeja

Academic Editor

PLOS ONE

Additional Editor Comments (optional):

Reviewers' comments:

Reviewer's Responses to Questions

**Comments to the Author**

1. If the authors have adequately addressed your comments raised in a previous round of review and you feel that this manuscript is now acceptable for publication, you may indicate that here to bypass the “Comments to the Author” section, enter your conflict of interest statement in the “Confidential to Editor” section, and submit your "Accept" recommendation.

Reviewer #1: All comments have been addressed

Reviewer #2: All comments have been addressed

2. Is the manuscript technically sound, and do the data support the conclusions?

Reviewer #1: Yes

Reviewer #2: Yes

3. Has the statistical analysis been performed appropriately and rigorously? 

Reviewer #1: Yes

Reviewer #2: Yes

4. Have the authors made all data underlying the findings in their manuscript fully available?

Reviewer #1: Yes

Reviewer #2: Yes

5. Is the manuscript presented in an intelligible fashion and written in standard English?

Reviewer #1: Yes

Reviewer #2: Yes

6. Review Comments to the Author

Reviewer #1: the authors have addressed all the comments and the manuscript is suitable for publication. The authors have added important data and figures

Reviewer #2: The authors have addressed almost all the concerns from the previous review cycle and the revised manuscript has been significantly improved.

7. PLOS authors have the option to publish the peer review history of their article (what does this mean?). If published, this will include your full peer review and any attached files.

Reviewer #1: No

Reviewer #2: No

---

## [Editor Report · Acceptance letter]

26 Feb 2021

PONE-D-20-20572R1 

Characterization of biliary microbiota dysbiosis in extrahepatic cholangiocarcinoma 

Dear Dr. Saab:

I'm pleased to inform you that your manuscript has been deemed suitable for publication in PLOS ONE. Congratulations! Your manuscript is now with our production department. 

Kind regards, 

on behalf of

Dr. Pradeep Dudeja 

Academic Editor

PLOS ONE